



# Assessing volumetric change distributions and scaling relations of retrogressive thaw slumps across the Arctic

Philipp Bernhard[1], Simon Zwieback[2], Nora Bergner[1], and Irena Hajnsek[1,3]

[1]Institute of Environmental Engineering, ETH Zurich, 8093 Zurich, Switzerland ETH Zürich
[2]Geophysical Institute, University of Alaska Fairbanks, Fairbanks, AK 99775 USA
[3]Microwaves and Radar Institute, German Aerospace Center (DLR) e.V., 82234 Wessling, Germany

**Correspondence:** Philipp Bernhard (bernhard@ifu.baug.ethz.ch)

**Abstract.** Arctic ice-rich permafrost is becoming increasingly vulnerable to terrain altering thermokarst, and among the most rapid and dramatic of these changes are retrogressive thaw slumps (RTS). They initiate when ice-rich soils are exposed and thaw, leading to the formation of a steep headwall which retreats during the summer months. These impacts, the distribution and scaling laws governing RTS changes within and between regions are unknown. Using TanDEM-X-derived digital elevation models, we estimated RTS volume and area changes over a 5-year period. We contrasted 9 regions (Eurasia: 4, North America: 5), with a total size of 220,000 $\mathrm{km}^3$, and over that time all 1853 RTSs combined mobilized a total volume of $17 \cdot 10^6 \, \mathrm{m}^3\mathrm{yr}^{-1}$ corresponding to a volumetric change density of $77 \, \mathrm{m}^3\mathrm{yr}^{-1}\mathrm{km}^{-2}$. Our remote sensing data revealed inter-regional differences in mobilized volumes, scaling laws and terrain controls. The area-to-volume scaling could be well described by a power law with an exponent of 1.15 across all regions, however the individual regions had scaling exponents ranging from 1.05 to 1.37 indicating that regional characteristics need to be taken into account when estimating RTS volumetric change from area change. The distributions of RTS area and volumetric change rates followed an inverse gamma function with a distinct peak and an exponential decrease for the largest RTSs. We found that distributions in the high Arctic were shifted towards larger values. Among the terrain controls on RTS distributions that we examined, slope, adjacency to waterbodies and aspect, the latter showed the greatest, but regionally variable association with thaw slump occurrence. Accounting for the observed regional differences in volumetric change distributions, scaling relations and terrain controls may enhance the modelling and monitoring of Arctic carbon, nutrient and sediment cycles.



## 1 Introduction

About one-quarter of the landmass in the northern Hemisphere is underlined by permafrost. With climate warming these permafrost regions become increasingly vulnerable to rapid thaw (Grosse et al., 2011; Schuur et al., 2015). Rapid permafrost

degradation has major impacts by changing ecosystem and hydrological equilibria. Furthermore it can impact the Earth system on a global scale by reinforcing climate change with the additional mobilization of organic carbon that was previously stored in the frozen soil. One important land surface characteristic arising from rapid thaw are retrogressive thaw slumps (RTS). These slumps initiate by the exposure of ice-rich soils with subsequent thaw and the formation of a steep headwall (Burn and Lewkowicz, 1990; Kokelj et al., 2009). During the summer, the ice in the headwall melts and leads to a continuous retreat. This

process can mobilizes vast quantities of sediments on a short time scale. In the context of recent warming an increase in the rates and sizes of RTSs in permafrost regions has been found (Lantz and Kokelj, 2008; Lantuit and Pollard, 2008; Gooseff et al., 2009; Kokelj et al., 2009; Lewkowicz and Way, 2019). However, the inter-regional differences in the rates of thaw slumping in terms of their magnitude, distribution and controls remain poorly constrained and so are the implications for carbon and nutrition cycles.

For the investigation of landslides in temperate climate zones, scaling laws of various form have been used to quantify hazards and ecosystem impacts as well as to improve the process understanding of landslide activity (Tebbens, 2020). The variability and similarities of these scaling laws in terms of landslides properties and area characteristics have played an important role. The soil properties (ice-content) as well as timescales (single event vs. polycyclic multi-year retreat) are different for RTSs than other landslides, but nevertheless the methods used as well as the universality of some scaling laws could provide valuable

insides into RTS drivers and controls. Furthermore, due to the strong spatial variability of soil-carbon densities as well as RTS activity past model estimates of the impacts of RTSs on the carbon cycle have large uncertainties (Turetsky et al., 2020). Quantifying the RTS scaling laws and their variability across regions have the potential to greatly improve future carbon release rates.

The two most important types of scaling laws are the frequency distribution as well as the area-to-volume scaling. For the

frequency distribution the area (or volume) change of the disturbed area showing elevation loss is commonly used. In this distribution typically two parts can be distinguished, an exponential decay part describing larger landslides and a deviation from this power-law for smaller events with a distinct peak, indicating the most common landslides in the region. The exponential decay part is well explained by models that merge closely proximal landslides. The attribution of the deviation from the power law is more controversial and is either attributed to an under-sampling of small events or to real physical processes (Tanyaş

et al., 2018). The second scaling law, namely the area-to-volume scaling, is based on an observational relation between landslide area and volumetric change. Many studies of landslides inventories that include different sizes, slope failure mechanisms and locations show that area-to-volume scaling follows a power law relation $V \propto A^{\gamma}$ with $\gamma$ ranging from 1 to 1.5. (Larsen et al., 2010). In a pure mathematical sense, a $\gamma$ of 1.5 corresponds to a situation were objects scale in an invariant way, meaning that if the height dimension is increased by a certain amount, the vertical (area) dimension is increased by the same. Consequently

a scaling coefficient smaller than 1.5 corresponds to a situation were an increase in area leads to a smaller but proportional





increase in height (Klar et al., 2011). The ability to estimate the volumetric change from area measurements can especially be useful for estimating the amount of mobilized material if only area measurement are available. Additionally, differences in $\gamma$ between regions may suggest different physical drivers of RTS development.

To quantify these relations for Arctic RTSs, only remote sensing techniques are feasible due to the remote landscape and the severe climate conditions. Digital elevation models (DEMs) that cover supra-regional the arctic permafrost terrain with a high enough resolution to study RTSs became only available in the last few years. One of these high resolution DEMs is based on single-pass InSAR observations taken by the TanDEM-X satellites. TanDEM-X is a high-resolution single-pass interferometry satellite mission that was launched by the German Aerospace Center (DLR) with the purpose of generating a high resolution global DEM (Krieger et al., 2007). The satellite pair started observations in 2010 and observed the global land areas two to three times. The expected spatial resolution of about $10\text{-}12\,\text{m}$ and vertical height resolutions of the order of about $2\text{-}3\text{m}$ is smaller than typical RTS change rates and can thus provide accurate estimates of the thaw slump topography as well as related controls on RTS processes like aspect, slope, and location (Bernhard et al., 2020).

In this study we use DEMs generated from TanDEM-X observations to derive the volume and area change rates of RTSs of several Arctic regions. Additionally we derive several terrain controls namely the aspect, slope, and location. This work focuses on answering the following questions:

1. Does the area and volume change probability density function of RTS follow the typical landslide distribution and to what extent does the function vary across regions?

2. What are the area-to-volume scaling law coefficients for the study regions and are they different?

3. Do the terrain controls vary between regions, and if so is the variation related to thaw slumps size?

The large number of thaw slumps in our sample and the diverse nature of our study regions allow for a robust statistical inference in answering these questions. The results should provide valuable insights concerning susceptibility modelling and to further improve our understanding of the process that govern RTS initiation and growth as well as their future impact.

## 2 Study Regions

We chose 10 different study sites located in permafrost regions across the Arctic Figure 1. The selection was first based on regions where previous studies have shown RTS activity: Peel Plateau and Richardson Mountains ("Peel"), Banks Island ("Banks"), western Mackenzie River Delta uplands and Tuktoyaktuk Coastlands ("Tuktoyaktuk") and Ellesmere Island ("Ellesmere") in northern Canada, Noatak Basin ("Noatak") in Alaska and Yamal and Gydan in Siberia (Lacelle et al., 2010; Balser et al., 2014; Segal et al., 2016; Nitze et al., 2017; Jones et al., 2019; Nesterova et al., 2019). Additionally, we chose three study regions in Siberia that exhibit RTS activity but are not well studied, namely on the Taymyr peninsula ("Taymyr 1 and 2") and on the Chuktoka peninsula ("Chukotka").

The study regions are located in the Arctic tundra and the boreal-tundra transition regions within the continuous permafrost



zone (Brown et al., 2002). They show differences in environmental properties including permafrost type, topography, lake-abundance and vegetation type. Due to the large size of some areas we selected only parts as our study region. The exact outline of the study regions were based on the Sentinel-2 tiling to facilitate the data processing steps.

The amount of ground ice on a pan-arctic scale has not been well characterized, but estimations on coarse scale report ground ice contents of >10% for all study regions (Brown et al., 2002). On small scales ground ice content varies strongly due to for example topography or soil type (Lacelle et al., 2004). The study regions on Peel, Banks, Ellesmere, Noatak and Chukotka show strong variation in topography with elevation changes of several hundred meters inside the study regions. The remaining regions show only small variation in elevation (<100m). Another difference between the study regions is in the amount of

lakes present. The study region with the most abundance of lakes are in Tuktoyaktuk and Taymyr 2. Only small amounts of lakes are found on Ellesmere and in Noatak (Table 1).

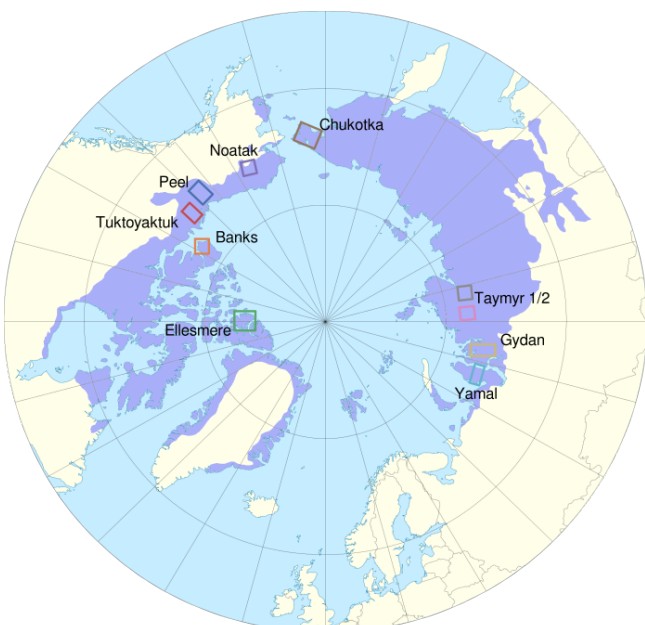

**Figure 1.** Overview of the study regions. The study regions are distributed around the Arctic with four study regions in northern Canada, one located in Alaska and five in Siberia. The purple area shows the continuous permafrost region (Brown et al., 2002).



| Study Regions | Size [$10^3$km$^2$] | lake area | Elevation [m] | TanDEM-X obs. [N] |
|---|---|---|---|---|
| Peel | 19.3 | 4.3% | 100 - 1500 | 307 |
| Banks | 6.6 | 6.3% | 0 - 400 | 62 |
| Tuktoyaktuk | 7.7 | 14.7% | < 100 | 87 |
| Ellesmere | 9.5 | 2.2% | 0 - 650 | 164 |
| Noatak | 16.3 | 1.5% | 400 - 1400 | 134 |
| Yamal | 24.8 | 6.0% | < 100 | 143 |
| Gydan | 14.6 | 8.9% | < 100 | 87 |
| Taymyr 1 | 23.6 | 4.1% | < 100 | 128 |
| Taymyr 2 | 11.0 | 11.1 % | < 100 | 124 |
| Chukotka | 87.9 | 1.4% | 0 - 1100 | 262 |

**Table 1.** Overview of study regions with size, lake area percentage, elevation range and number of processed TanDEM-X observations. The Lake area percentage was calculated using the generated waterbody mask. Open water bodies were not included in the calculation.



## 3 Methods

### 3.1 Data and Processing

For the DEM generation we used TanDEM-X observations acquired between 2010 and 2017. To ensure adequate vertical

accuracies, we only used acquisitions with a height of ambiguity smaller than 80 m (Martone et al., 2012). The incidence angles range from $36°$ - $44°$. For an accurate orthorectification we use the TanDEM-X 12m DEM as reference and iteratively updating the look-up table based on the measured deviation (Leinss and Bernhard, 2021). We only studied winter acquisitions since due to the low average monthly temperature we can expect a dry snow-pack and radar waves can propagate through without being strongly affected (Millan et al., 2015; Leinss and Bernhard, 2021). Outside the winter months thawed vegetation, wet snow

and standing water induce sizeable errors (Bernhard et al., 2020). For the DEM generation we followed a standard approach (Fritz et al., 2011). The resulting DEMs have a planimetric resolution of about $10$ - $12\,\mathrm{m}$ and vertical accuracies of about $2\,\mathrm{m}$ in areas with high coherences. The interferometric processing was done using the *Gamma Remote Sensing* software (Werner et al., 2000). More processing details including tilt-removal and correction of misalignments, specifically for DEMs generated from InSAR observations in permafrost regions can be found in Bernhard et al. (2020).

### 3.2 RTS detection and manual mapping of affected areas

DEMs corresponding to the same winter were averaged and mosaicked. We then used an automated detection algorithm to identify significant elevation changes in the DEM difference images from DEMs that were obtained more than 3 years apart (Bernhard et al., 2020). For each detection several processing steps were carried out. First the topography and environment were assessed using a TanDEM-X DEM and Sentinel-2 multispectral images taken in summer (snow-free). For all study regions at

least one Sentinel-2 image during the years 2016 to 2019 was available. The criteria for classifying a detection as an active RTS were the exposure of bare soils, a retreat over time, a location related to a potential sediment removal mechanism, and the presence of a headwall (Lantuit and Pollard, 2008; Nitze et al., 2018; Lewkowicz and Way, 2019). In uncertain cases additional time-series of Planet Rapid-Eye optical data was used to classify the detections (Planet-Team, 2018). At Banks a previous study showed that this method of detecting RTSs gives a false negative rate of 26% (Bernhard et al., 2020).

After the classification step we generated polygons for each detected RTS outlining the area with significant elevation change. An automated method using a fixed threshold on the elevation change gave unreliable results. Thus the polygons outlining the area of elevation change were drawn by a trained student and the first author. Additionally, the location of the RTS in terms of "shoreline" or "hillslope" was noted.

### 3.3 RTS attributes

For all calculations we used the area outlined by the polygon indicating the areas showing an elevation change. We computed the volumetric and area change as well as the slope and aspect. In some cases observations during several winters in 2010/11, 2011/12 and 2012/13 were available. To simplify the analysis we normalized the properties to changes per year and took the





average if several DEM difference pairs were available. We computed the aspect and slope by using the pre-disturbed elevation model and applied gaussian smoothing with a standard deviation corresponding to $100\,\mathrm{m}$ to reduce the influence of random

errors (Kang-tsung and Bor-wen, 1991). For the aspect distribution we additionally computed the aspect distribution weighted by volume.

To quantify the volumetric change rate density (volumetric change per unit area) we first use a simple approach by dividing the study region size by the total volumetric change rates. This has a drawback because RTSs often occur heterogeneously and the result strongly depends on the exact outline of the study region (Ramage et al., 2017). For example, in the Peel study

region only the east facing part of the mountain range experiences RTS development, but this study region also includes the western part of the range where nearly no RTS activity was detected. To account for this problem we follow a similar approach then proposed in Kokelj et al. (2017) and divide our study region into tiles of sizes $10\,\mathrm{km}\,\mathrm{x}\,10\,\mathrm{km}$ and counted the number of empty grid cells and computed a more representative RTS density using only the cells showing RTS activity. It is to note that to interpret the computed density values the number of empty as well as the number with non-empty grid cells in relation to

the total size of the study region should be considered.

To quantify the amount of lakes in each study region we used the waterbody mask generated from Sentinel-2 data and computed the area that is covered by the mask (McFeeters, 1996; Kaplan and Avdan, 2017). For this computation we excluded open sea areas.

We investigated the dependency of RTS growth on different terrain controls by computing the aspect, slope and the location

in terms of lakeshore- and hillslope-RTSs. For the aspect we identified the most dominant orientation by summing the number of RTSs as well as the volumetric/area change rates in 8 aspect bins (N, NE, E, SE, S, SW, W, NW) and used these bins to compute the strength and orientation of the primary direction.

### 3.4 Change Rate Distributions

The probability density function (PDF) of the area effected by elevation loss per year corresponding to an RTS inventory can

be defined as

$$\mathrm{p}(\mathrm{A_{RTS}}) = \frac{1}{\mathrm{N_{RTS}}}\frac{\delta \mathrm{N_{RTS}}}{\delta \mathrm{A_{RTS}}} \tag{1}$$

where $\mathrm{A_{RTS}}$ is the area change effected by elevation loss of an RTS per year, $\mathrm{N_{RTS}}$ the total number of RTS in the inventory, $\delta \mathrm{N_{RTS}}$ the number of RTS with affected areas between $\mathrm{A_{RTS}}$ and $\mathrm{A_{RTS}} + \delta A_{RTS}$ and $\delta \mathrm{A_{RTS}}$ is the bin width. Equivalently the probability density function $\mathrm{p}(\mathrm{V_{RTS}})$ for the volumetric change per year can be defined.

All RTSs in the study show changes per year in the range of $10^2$ to $10^6$ $\mathrm{m^2}$ $\mathrm{yr^{-1}}$ for area and $10^2$ to $10^6$ $\mathrm{m^3}$ $\mathrm{yr^{-1}}$ for volumen thus we used 30 bins sampled in log-space to cover these ranges.

When analysing a landslides PDF three quantities can be used to describe the distribution: the rollover- and cutoff-points for small events and the coefficient of the power law scaling $\beta$ for large events. The rollover point is defined as the peak in the PDF and corresponds to the most common occurrence in the distribution. For large slumps the PDF can be described as a power law

function. The point at which the distribution starts to follow a power law is defined as the cutoff point (Figure 2).

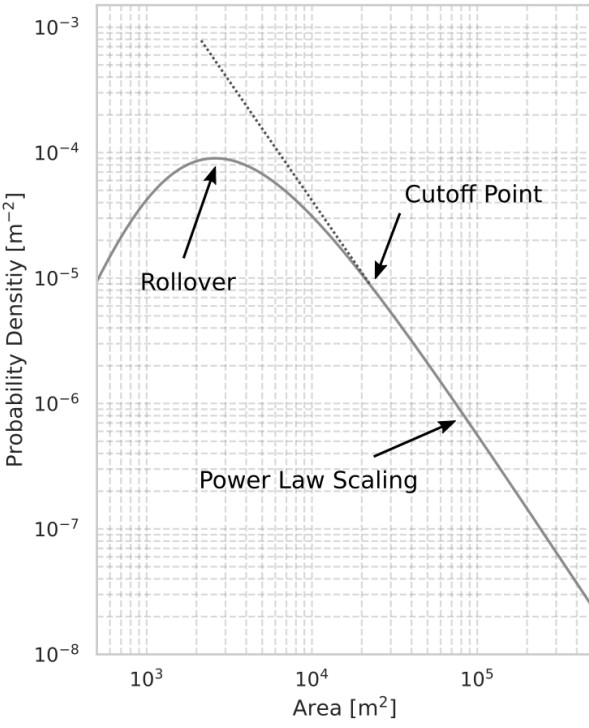

**Figure 2.** Schematic representation of RTS area probability density function. Two parts can be distinguished: An exponential decay part above the cutoff value and a deviation from the power-law scaling below the cutoff point.

To determine how well the data points are described by this model and to estimate the rollover point we fit a three parameter Inverse Gamma Function to the RTS probability density function (Malamud et al., 2004). To estimate the error of the fit we used the bootstrap method drawing 1000 random samples with replacement from all data points and for each iteration computed the $R^2$ value as well as the rollover point (Ohtani, 2000).

For the computation of the cutoff value and the exponential scaling exponent we use the method proposed in Clauset et al. (2009) which is commonly used in landslide frequency-area analyses (Bennett et al., 2012; Parker et al., 2015; Tanyaş et al., 2018). The approach is based on sampling all possible cutoff values and estimating corresponding exponential scaling coefficients $\beta$ using a maximum-likelihood fitting method. The obtained fitting values are then tested based on a Kolmogorov-Smirnov statistic and the values that follows best a true power law distribution are used as the final cutoff and $\beta$ value. To

quantify the uncertainty we again used a bootstrap algorithm.

### 3.5 Area-Volume scaling

One important quantity in comparing landslides of various sizes is the relation between volume and area. The simplest conversion assuming an anisotropic scaling with a scaling exponent $\gamma$ to relate the area and volume: $V \approx A^\gamma$. Since both variables





(area and volume) are affected by measurement errors we used an orthogonal distance regression model to fit a straight line
(Boggs and Rogers, 1990; Markovsky and Van Huffel, 2007). To quantify the goodness of the fit we calculated the RMSE, $R^2$
and p-value (in log-space).

## 4 Results

We investigated 10 different study regions and measured the area and volumetric change rates of 1854 RTSs over a 4 - 5 year
time-frame. Due to the low density of RTSs in Yamal and Gydan and the two study regions in Taymyr we combined these to
one study region (in the following "Yamal/Gydan" and "Taymyr") according to their geographical and geophysical proximity.
The number of RTSs per study region and the obtained volumetric change rates in terms of the total volume, the density and
the changes per RTS are shown in Table 2. The area and voluemtric change rate distribution as violine plot are shown in Figure
3. We found the largest RTSs in terms of average volumetric change rates per RTS on Ellesmere, Peel and Banks with yearly
average yearly change rates of $13200 \, \mathrm{m^3 \, yr^{-1}}$, $12200 \, \mathrm{m^3 \, yr^{-1}}$ and $10700 \, \mathrm{m^3 \, yr^{-1}}$. The other areas show much smaller yearly
average volumetric change rates in the the range of $2400 \, \mathrm{m^3 \, yr^{-1}}$ (Tuktoyaktuk) to $3600 \, \mathrm{m^3 \, yr^{-1}}$ (Taymyr). For the total
volumetric change and the volumetric change density of the study regions the same difference is visible with a separation of
one to two orders of magnitude.

In the following paragraphs we will present (1) a characterisation of the area and volumetric changes rates with special emphasis on the probability density functions with the estimation of the rollover, cutoff and exponetial decay components, (2) the
estimated area-to-volume scaling laws and (3) several terrain controls that could potential be related to RTS size and frequency.
To compare the estimated quantities in each section we computed the correlation coefficients between them. All coefficients
are shown in the Supplement Figure S1.

| **Area** | $N_{RTS}$ | $V_{change}^{total}$ $[10^6 \mathrm{m^3 \, yr^{-1}}]$ | $V_{change}^{mean}$ (density) $[\mathrm{m^3 \, yr^{-1} \, km^{-2}}]$ | $V_{change}^{mean}$ (RTS) $[10^3 \mathrm{m^3 \, yr^{-1} \, RTS^{-1}}]$ |
|---|---|---|---|---|
| Peel Plateau | 438 | 5.27 | 342.8 | 12.2 |
| Banks Island | 679 | 7.16 | 883.8 | 10.7 |
| Ellesmere Island | 223 | 2.95 | 546.7 | 13.2 |
| Tuktoyaktuk | 212 | 0.5 | 43.3 | 2.4 |
| Noatak | 26 | 0.09 | 14.9 | 3.4 |
| Yamal/Gydan | 128 | 0.37 | 12.4 | 2.9 |
| Taymyr | 97 | 0.35 | 11.3 | 3.7 |
| Chukotka peninsula | 51 | 0.17 | 3.8 | 3.5 |

**Table 2.** Number of RTSs in each study region with the total number of RTS and the volumetric change rates in terms of total change, density
and average rates per RTS.



## 4.1 RTS volume and area distributions

The estimated PDF's are shown in Figure 4 a and b. For most areas the quality of fit of the inverse gamma function was good,
as indicated by $R^2$ values $> 0.8$. Exceptions were the Noatak and Chukotka study regions with $R^2$ values between 0.6 and 0.7.
These two regions have also the lowest number of RTSs in the sample with only 26 (Notatak) and 51 (Chuktoka) RTSs.

The modes of the volume change distributions (rollover points) differ between regions. The two study regions located in the
high Arctic (Ellesmere and Banks Island) show an order of magnitude higher rollover values. The distribution of the area and
volumetric change rates in form a violin plot can be seen in Figure 4 c and d. The range of measured volumetric and area
change rates show large variations for the Tuktoyaktuk and Peel study region, whereas the other study regions show only small
variation.

The PDF's above the cutoff value, the relation between rollover and cutoff as well as the exponential decay values can be seen
in Figure 5. For the PDF based on the volumetric change, a high rollover value is moderately associated with high cutoff values
indicated by a correlation coefficient of 0.59 (Figure **??**). In contrary, the PDF based on the area change rate shows a much
stronger separation between the high Arctic regions and the other study regions and consequently also shows a high correlation
factor of 0.98. For the power law exponent for RTSs above the cutoff values no large difference between the areas is visible
($\beta \approx 2$ to $3$ and correlation coefficients $< 0.64$). It is to note that for the yearly area and volumetric change rates the cutoff value
for the Peel study region is relatively small but the distribution continues to high values with yearly area change rates of up to
$2 \cdot 10^5 \mathrm{m}^2 \mathrm{yr}^{-1}$ and $1 \cdot 10^6 \mathrm{m}^3 \mathrm{yr}^{-1}$ (Mega-Slumps). The computed values of the rollover, cutoff, exponetial decay coefficients
as well as the fit parameters for the inverse gamma function are reported in the Supplement Table S1 and S2.

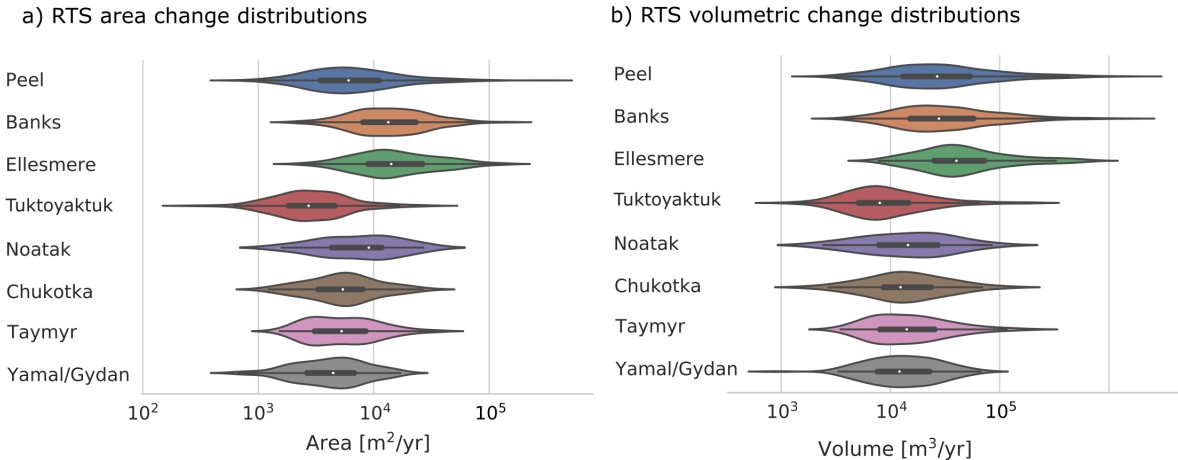

**Figure 3.** Area and volumetric change rate distributions of mapped RTSs. a) and b) show the distribution of area and volumetric change rates
in form of violin plots. The white dots on the center lines indicate mean values.





**Figure 4.** PDF of area and volumetric change rates of mapped RTSs. a) and b) shows the PDF of area and volumetric change rate with fitted inverse gamma function. c) and d) show the computed $R^2$ errors.



**Figure 5.** Cutoff, rollover, and exponential decay coefficents. a) and b) shows the PDFs for yearly area and volumetric change rates above the cut-off values. c) and d) shows the estimated rollover and cutoff values for yearly area and volumetric change rates. e) Exponential decay coefficients for fits above the cutoff.





## 4.2 Area-to-volume scaling

The estimated area-to-volume scaling law based on all data points in log-log coordinates can be seen in Figure 6 a. A clear relationship that spans over four order of magnitudes between the area and volumetric change rates is visible. The estimated scaling exponent across all regions was $\alpha = 1.15 \pm 0.01$. The quality of fit was decent, with a $R^2$ value of 0.81, RMSE of 210   $0.21 \, \mathrm{m^3/yr}$ and p-value smaller than $10^{-6}$ showing a strong dependency between RTS area and volumetric change rates. This is remarkable considering that RTSs in the sample occurred in different topographic and geomorphological settings. Neverthe-less we found a moderate inter-region variability in the scaling coefficient. The $\alpha$ coefficients for the individual regions is in the range from 1.05 to 1.25 with the exception of RTSs in the Banks region with a high coefficient of 1.37. The datapoints and fitted lines for each study region can be seen in the Supplement Figure S2. The strong association between area and volume 215   scale rates can facilitate the estimation of volume changes from multispectral satellite images.

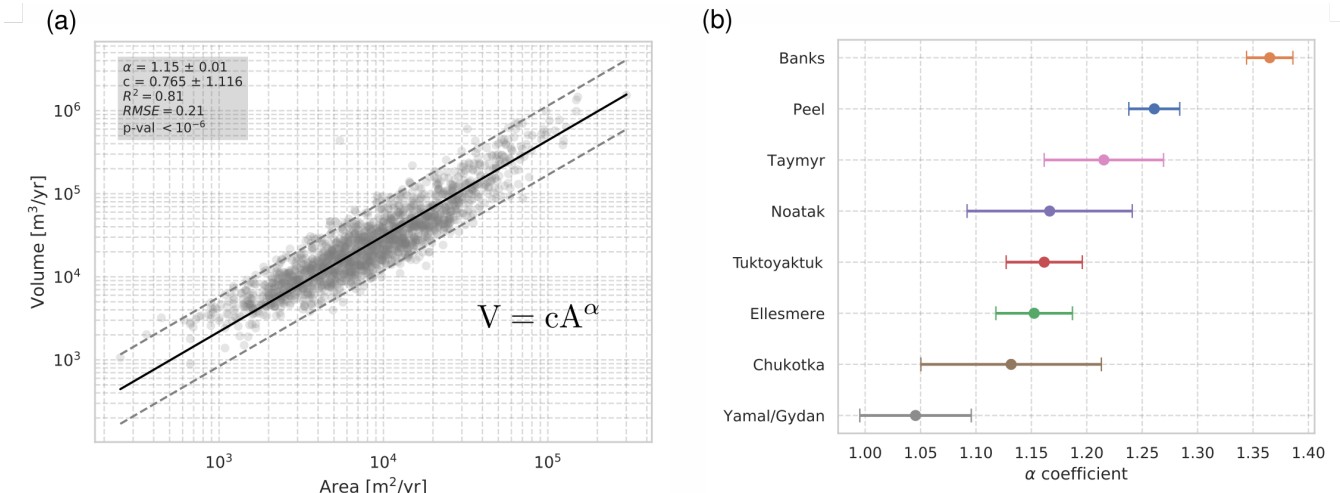

**Figure 6.** Area-to-volume scaling laws. (a) shows the total dataset with all regions combined. We found an exponential scaling exponent of $\alpha = 1.15 \pm 0.01$. (b) shows the computed values of the scaling exponent for each region individually. A large variation between 1.05 and 1.37 is visible.





### 4.3 Terrain controls

Among the investigated terrain controls, aspect shows the greatest difference between region. RTSs located in the study regions in Siberia as well as on Ellemere Island tend to favour a South-West facing orientation (Figure 7a). The very small number
of RTSs in the Noatak study region showed a preferred orientation towards the North-West and RTSs in Peel have a preferred orientation towards the North-East. For Tuktoyaktuk and Banks no clear trend is visible. To consider the possibility of more than one preferred orientation we additionally looked at the initial aspect bin distribution (see Supplement Figure S3). Here only the aspect distribution of RTSs in the Noatak valley shows two preferred orientation, but this could be related to the low number of RTSs in the study region. Additionally to the number of RTSs in each aspect bin we weighted the aspect by
the volumetric change rates. This does only slightly alter the preferred orientation and large slumps do not occur at different aspects than the typical RTSs.

The slope of the pre-disturbed area shows some difference between the study regions (Figure 7c). In general all RTSs evolve at slopes ranging from 2°-3° up to slopes of 20°. Interestingly, in the region of the largest slumps on Banks Island, RTSs tend to favour lower slopes with values below 12° .

We investigated the dependency of RTSs locations in terms of their occurrence. We distinguished two types of locations, either at a shore (including lake and coastal) or at hillslopes with no large waterbodies close by. Several regions have mostly one type of RTS location. The RTSs in Ellesmere (99% hillslope), Peel (96% hillslope) and Selawik (88% hillslope) have mostly RTSs at hillslope locations. On the contrary, RTS in the Tuktoyaktuk region has most RTSs at lakeshores (99%). All other regions have a mixture of hillslope and shoreline RTS locations: Banks (66% hillslope), Chukotka (52% hillslope), Taymyr
(27% hillslope) and Yamal/Gydan (26% hillslope). In the regions with both types of RTS locations no significant difference between the distributions is visible (Figure 7c). Furthermore, we did not find a significant correlation between RTS size and the percentage of hillslope or shoreline RTS locations (Supplement Figure S1).

To estimate the volumetric change rate density of RTSs within the RTS-affected regions of each study region we gridded them into tiles of size 10km · 10km. Using only the tiles with RTSs present, Figure 8a shows the volumetric change rates per square
kilometre. The volumetric change densities over the total study region (as shown in Table 2) strongly depends on the exact outline of study regions and removing tiles without RTSs present should give a more consistent and comparable volumetric change rate density. To make this more visible the amount of tiles with RTSs present and without can be seen in Figure 8b. For example on the Chukotka pensinsula only a small number of tiles have active RTSs present.



**Figure 7.** Terrain controls of mapped RTSs for each region. (a) shows the aspect main orientation of RTSs in each region (left) and additionally weighted by the volumetric change rates (right). (b) shows the probability density distributions of volumetric changes rates separated by RTS location. On the rights side the Number of RTS in each subsample is listed. Some areas are dominated by one location type. (c) shows the distribution of the pre-disturbed DEM slopes at the RTS locations.


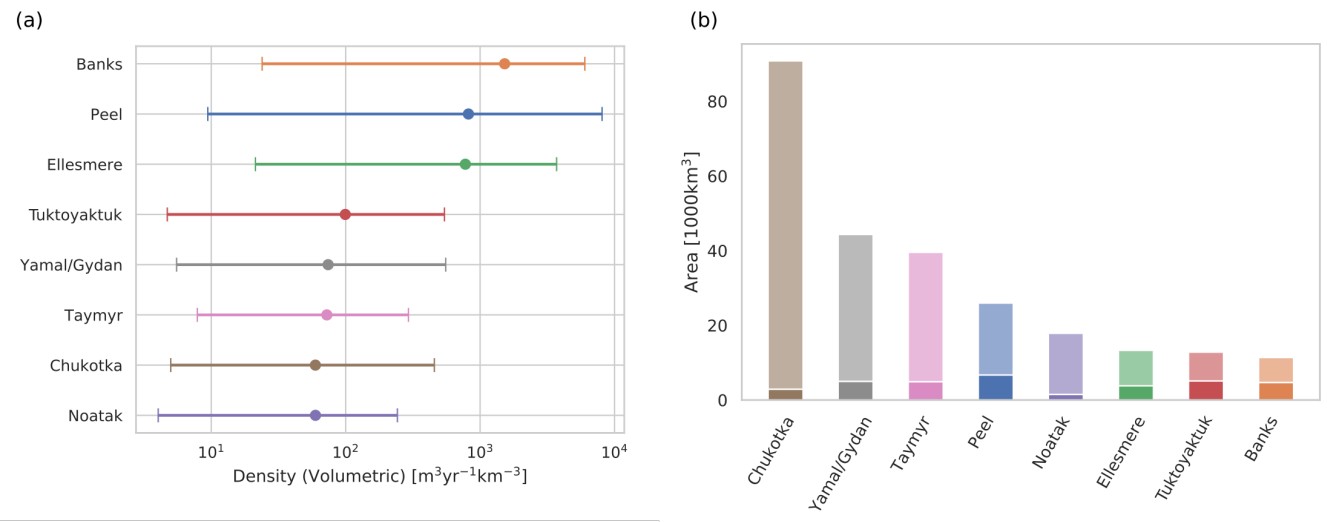

**Figure 8.** (a) shows the computed RTS volumetric change rate densities using a 10 km by 10 km grid with the empty grid cells removed. The vertical bars indicate the range in the computed densities. (b) shows the study region size in transparent with the fraction of tiles with RTS present in solid (b).





## 5 Discussion

### 5.1 Probability density functions to characterize thaw slump activity

The computed probability density functions of the yearly area and volumetric change rates follow a characteristic inverse gamma law with first an increase in frequency up to a maximum value with the most abundant thaw slump sizes (rollover) and then a decrease with an exponential decay tail above a certain cutoff value. Our findings show that the applicability of this universal scaling also pertains permafrost landscapes, despite differences in the governing geomorphic processes with respect to lower latitude environments. To further investigate the distributions we distinguish between two parts: (1) the exponential decay part for large RTSs and (2) the part that deviate from this exponential decay below the cutoff point. For landslides the exponential decay part is typically explained in a statistical way by the concept of self-organized criticality, where a constant "input" of a specific landslide size at random location, together with a merger of landslides that are close to each other, reproduces this distribution (Bak and Tang, 1989; Turcotte, 1999). For RTSs this explanation seems plausible since initiation and evolution are strongly linked to soil properties that can promote RTS development in close proximity and also RTS coalescence is common (Lantz and Kokelj, 2008; Lantuit et al., 2012; Wang et al., 2016). In addition to the universal exponential decay behaviour in all study regions we found that the largest RTSs in the Peel, Banks and Ellesmere study regions have order of magnitudes larger growth rates (Figure 5 a,b). A possible explanation is that topographic and geomorphological properties, like the amount of relic ice, overburden thickness or the steepness of terrain only allow RTSs to grow to a certain size (Kokelj et al., 2017; Rudy et al., 2017; Jones et al., 2019). For example in the Tuktoyaktuk study region ($N_{RTS} = 212$) were RTSs occur at lakeshores in mainly flat regions, the largest RTSs show growth rates of $5200\,\mathrm{m^2yr^{-1}}$ and $31800\,\mathrm{m^3yr^{-1}}$ compared to for example the Ellesmere region ($N_{RTS} = 223$) with more topographic features and mainly hillslope RTSs which shows 3-4 times higher maximum growth rates ($23000\,\mathrm{m^2yr^{-1}}$ and $106400\,\mathrm{m^2yr^{-1}}$). This suggests that additional to the exponential decay factor also a maximum RTS growth rate is important to characterize the high end tail of the probability density function.

For the deviation from the exponential decay, two types of explanation have been proposed for landslides in temperate climate (Tebbens, 2020). First an under-sampling of small landslides due to limitations in resolution and secondly an explanation that attributes this divergence on physical processes. By investigating our dataset a divergence due to under-sampling seems unlikely since the areas in Peel, Banks and, Ellesemere show this divergence (cutoff-point) at high yearly change rates of $> 10^4\,\mathrm{m^2yr^{-1}}$ and $> 3{\cdot}10^4\,\mathrm{m^3yr^{-1}}$ which corresponds to area and volumetric changes high above the resolution limit (TanDEM-X resolution: Spatial $\approx 10\,\mathrm{m}$, vertical $2-5\,\mathrm{m}$). The physical origins are likely related to environmental conditions and soil properties like ground ice-content but are outside the scope of this work. Future models for thaw slump initiation and evolution should be able to investigate the drivers and reproduce such distributions.

### 5.2 Similarities and differences in Area-Volume scaling

We found a power law relationship ($V \approx A^\alpha$) between the area and the volumetric change rates with a scaling coefficients $\alpha$ of 1.15 for the total dataset and ranging between 1.05 and 1.37 for the individual regions. Such relationship are known from





landslides in temperate climates with typically values of 1 to 1.5 (Larsen et al., 2010; Klar et al., 2011). For RTSs only one study by Kokelj et al. (2020), investigating thaw slumps on the Peel Plateau and Richardson Mountains, has estimated this relationship and found a scaling coefficient of 1.42 which is relatively high compared to our values (Peel: 1.27, Tuktoyaktuk:
1.17) but inside the estimated error.

Comparing the coefficients between regions we found that lower scaling coefficients are not correlated with smaller slumps. For example the scaling law coefficient in the Tuktoyaktuk region with relatively small slumps is the same as for the RTSs in the Ellesmere region with the largest slumps in our dataset. Further investigations relating the scaling coefficients to additional RTS and area characteristics (e.g. soil properties, climatic history) are needed.


## 5.3   Terrain controls and their relation to RTS size

With the available data we could determine several terrain controls, namely the orientation of thaw slump growth, the slope of the predisturbed area the slumps grew into as well as the location in terms of hillslope and shoreline RTSs. Our findings in terms of the preferred orientation of RTSs are mostly consistent with past regional studies: A preferred South-West orientation
for RTSs in the Siberian study regions (Nesterova et al., 2019) and Ellesmere Island (Jones et al., 2019), towards the North-East for the Peel Plateau study region (Lacelle et al., 2015) and North facing RTSs in the Noatak-Valley (Swanson and Nolan, 2018). For RTSs in the Tuktoyaktuk region we found no preferred orientation consistent with Wang et al. (2009), but in contradiction to other studies that found RTSs orientations that favour North facing slopes (Kokelj et al., 2009; Zwieback et al., 2018, 2020). The association with aspect hints at inter-regional differences in the governing geomorphic drivers and controls. A South-West
facing orientation is considered to be related to higher initiation- and growth-rates of RTSs due to the higher energy available from solar radiation (Lewkowicz, 1987). This would suggest that solar radiation is an important factor in RTS growth and initiation for the study regions in Ellesmere and Siberia. Past studies have shown that a high ground ice content is a necessary condition for RTS development (Kokelj and Jorgenson, 2013; Ramage et al., 2017). During the Holocene Thermal Maximum the regions in North-West Canada experienced warmer summer temperatures than other arctic regions and could have removed
ground ice on South-facing slopes (Burn et al., 1986; Kaufman et al., 2004; Lacelle et al., 2010; Zwieback et al., 2018). Thus the differences in RTS aspect distributions could be related to the climatic history. For example the dominant north-facing exposure on the Peel Plateau could reflect such anisotropic abundance of ground ice.

We did not find a significant relation between thaw slump size (area and volumetric change rates) to aspect as well as slope and location (hillslope, shoreline). This finding affirms previous studies that highlighted the complexity of the processes and
controls governing thaw slump expansion.

## 5.4   Implications

The scaling relations we quantified are critical for modelling and predicting thaw slumping and its impact on biogeochemical cycling. The regional variability in scaling behaviour needs to be considered when upscaling field observations to estimate large-scale nutrient, sediment, and carbon budgets. Because Earth system models strive to capture the variability of these



processes from regional to global scales, our results can be used to calibrate and validate global models. Possible changes in the scaling relations could be important indicators to predict future thaw slump evolution and impacts.

Our observations of variable thaw slump rates and regimes highlight the need for continual pan-Arctic monitoring and further satellite missions to derive high resolution DEMs. The TanDEM-X data availability only allowed to compute elevation changes in a 4 to 5 year time window. To investigate changes in thaw slump activity related to climate change a higher temporal resolution is needed. Here additional observations from the TanDEM-X satellite as well as data from the ArcticDEM could add additional datapoints (Bachmann et al., 2018; Dai et al., 2020). Furthermore, with the derived area-to-volume scaling laws it is potentially possible to use optical satellite images which are available at a higher temporal resolution to estimate the volumetric change.

## 6 Conclusions

In this study we quantified the yearly volumetric and area change rate of RTSs over a 4-5 year time-frame in 10 study regions across the continuous permafrost of the Arctic with a total study size of 225100 $\text{km}^2$ and a total number of 1868 RTSs. We found that the frequency distributions of the volumetric and area change rates are well described by an Inverse Gamma distribution ($\text{R}^2 > 0.5$) with the distinct features of a rollover, cutoff and an exponential decay for large RTSs. This kind of behaviour is well known for landslides in temperate climate regions with very different trigger mechanisms and soil properties and could provide valuable insides in modelling future RTS evolution on a pan-Arctic scale.

The comparison between regions showed that the distribution of RTSs in three study regions in northern Canada (Peel Plateau and Richardson Mountains, Banks Island, Ellesmere Island) are shifted towards higher change rates in volume and area. Nevertheless, the exponential decay rates for large RTSs in all regions were similar.

Our analyses revealed consistent but regionally variable area-to-volume scaling behavior. For the total dataset we found a scaling coefficient of $\gamma = 1.15 \pm 0.01$ with some variance between the study regions ($\gamma = 1.05 - 1.37$).

For the aspect we found diverse preferred orientations of RTSs between the study regions from no dominant orientation for Tuktoyaktuk and Banks, a NE orientation for Peel, E-facing RTS in the Notatk valley and a strong SW orientation of all study regions in Siberia and the study region in Ellesmere.

Our regionally variable thaw slump scaling relations may be used to constrain large-scale estimates of carbon, sediment and nutrient budgets. By capturing the variability of thaw slump rates across scales, remote sensing is a vital tool for predicting hazards and attendant ecosystem changes in a rapidly changing Arctic.

*Data availability.* Locations and extracted properties of RTSs are available at: https://www.doi.org/10.3929/ethz-b-000482449. The polygons outlining the area of elevation change are available upon request. Sentinel-2 are available from the Copernicus Open Access Hub (https://scihub.copernicus.eu). TanDEM-X CoSSC data are not freely available but can be requested from the German Aerospace Center (DLR) and accessed through the EOWEB (https://eoweb.dlr.de)





*Author contributions.* PB conducted the DEM processing, analysed the data and drafted the initial manuscript, SZ provided critical guidance and contributed to the writing of the manuscript, NB and PB conducted the manual RTS mapping, IH provided guidance and corrections to the final manuscript.

*Competing interests.* The authors declare no conflict of interest



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
