# Peer review of "Assessing volumetric change distributions and scaling relations of retrogressive thaw slumps across the Arctic"

_The Cryosphere, 2021_

## Author Comment (AC2)

[Figure]

**Figure 1.** Example of RTS polygones in the Chukotka study region located at N65.93 W-178.82. Left: DEM difference image between winter 2010/11 and 2016/17. Right: False color Sentinel-2 image taken on the 09.09.2016.

[Figure]

**Figure 2.** Example of RTS polygones in the Tuktoyaktuk study region located at N69.08 W-134.02. The black arrows indicate the RTS lcoations Left: DEM difference image between winter 2010/11 and 2016/17. Right: False color Sentinel-2 image taken on the 06.08.2016.

[Figure]

**Figure 3.** Example of RTS polygones in the Peel study region located at N67.26 W-135.27. Left: DEM difference image between winter 2010/11 and 2016/17. Right: False color Sentinel-2 image taken on the 06.08.2016.

[Figure]

**Figure 4.** Example of RTS polygones in the Yamal study region located at N71.09 W70.40 . Left: DEM difference image between winter 2010/11 and 2016/17. Right: Fasle color Sentinel-2 image taken on the 12.08.2016.

---

## Author Response (AR1)

We thank the reviewer for the detailed and constructive comments. We considered each comment carefully and address them point by point below. We hope that our modifications will make the manuscript clearer and more complete.

In the following we abbreviate: RC1 (Reviewer 1 Comment), RC2 (Reviewer 2 Comment) and
5   AC (Author Comment)

**Reviewer 1**

**RC1 paragraph 2**

The analysis focuses first on thaw slump volume, the average volume change among the three change models potentially available (ln 122-123). It is important to note in this paper that thaw
10   slumps are chronic, multi-year (often multi-decade) features that produce variable eroded volume over time, and the erosional intensity and morphological complexity tends to change with the age of the feature. This is a critical point of distinction from landslide studies that commonly examine the scaling of the total erosion from a scar zone. Both approaches have important outcomes: from an annual yield perspective, the area-volume scaling relationships presented here agree well with
15   established power law parameters, and the resultant regression will likely be helpful for estimating annual yield from mapped RTS polygon areas. From the perspective of the full scar erosion depth, measures of the time-integrated changes in morphology can yield a regression that might tell us more about the longer-term trajectory of the landscape. However, generating a 'pre-disturbance' surface can be time consuming, and there is the prospect of erroneous reconstruction, particularly
20   in the case of larger slumps in complex topography. I'm not suggesting the latter analysis be incorporated, but it is important to highlight this distinction.

**AC**

We agree that this distinction is an important point to make and we edited the manuscript at two points to highlight this distinctions:
25   In Methods-Section 3.3 we added:
"It is to note that RTS are multi-year features with a strong variability in the erosional intensity as well as a potential change of their morphology over time. In the interpretation of the results and specifically the comparison to landslide studies the use of the integrated change over several years needs to be considered."
30   in the Discussion-Section 5.1 we added:
"It should be emphasized here that another difference of our analyses to common landslides studies is that RTSs are a multi-year phenomena with variable yearly erosion rates. Some variability in the exact form of the distributions should therefore be expected if different time periods are chosen."

**RC1 paragraph 3**

The authors should perhaps clarify that the "area" term denotes primary scar zones (only) - not including spoil zone or other reworking, for clearer comparison with other datasets. There is invariably some detritus that fills the primary erosion site, particularly in older and larger RTS features on more subdued slopes, so the precise volume of most recent erosion is not always accessible.

**AC**

To clarify the term "area" is this context we added to the Section 3.3 RTS attibutes the sentences: "For all calculations we used the area outlined by the polygon indicating the areas showing an elevation change and thus a net volume loss. It is to note that this area can also be a zone of deposition, especially for small and low-relief RTSs or if the time between observations increases. Areas such as the debris tongues or zones of alluvial deposits can not be accurately detected by the DEM difference data and are not included."

**RC1 paragraph 4**

The term "volumetric change rate density" (ln 127) is clarified as "volumetric change per unit area", but the statement goes on to say this is calculated "by dividing the study region size by the total volumetric change rates", which seems to be rather the reciprocal – and a "change rate" (e.g. ln 275) is different from volumetric change. I'm perhaps misunderstanding your intent here, but some clarification of this specific yield term is needed.

**AC**

We introduced the term "volumetric change rate density" to investigate how much volume is eroded in a specific area (analogous to a "RTS density" - the number of RTS found in an area e.g. a square kilometre). We made a mistake in saying that the "volumetric change rate density (volumetric change per unit area)" was computed by dividing the study region size by the total volumetric changes per year. We added some clarifications and the correct sentence is: "To quantify the volumetric change rate density (volumetric change rate per unit area) we first use a simple approach by dividing the summed total of all RTS volumetric changes per year by the study region size." We hope that with this mistake fixed is is clear what we are trying to do.

**RC1 paragraph 5**

While the TanDEM-X elevation dataset has broad statistical characterization of the vertical accuracy (ln 101-102), the problem of volumetric change in landslides, gullies and other mass-wasting zones present a more specific problem: how well is the the scar zone volume characterized by the

grid of elevation values interpolated in and around it? Given the focus on allometric relationships, it is important to assess the propagation of various errors, some that are likely to vary with scale. As the scale of erosion features approaches the pixel resolution, the estimated volume will be increasingly approximative. Admittedly the problem of error characterization and propagation in landslide inventories has not advanced very far generally, but given that this work could be a stepping stone to even further extrapolations of sediment and carbon export, it would be quite helpful to establish a list of factors that contribute to error and some estimation of the overall precision that can be achieved with this methodology. Some calibration with finer-scale elevation datasets could help with this problem, as well.

**AC**

We agree with the reviewer that a detailed quantification of errors would be very helpful, unfortunately quantifying the error in a rigorous way is very challenging due to the combination of a spatial, a vertical as well as a time component that is necessary to characterize a RTS. To investigate this in more detail a reference dataset with a high resolution change estimates with a sufficiently closely matching temporal period. This is currently not available. We addressed this problem in greater depth in your last publication ((Bernhard et al., 2020)) To include a more detailed error estimation and uncertainties we added the following to the Method section 3.2:
"The lower limit for a RTSs to be detectable in terms of headwall height and retreat is very hard to quantify due to the limited amount of available high resolution, three dimensional RTS inventories. Here also the timescales on which the RTSs are monitored plays an important role. The 90th percentile in terms of elevation changes of the 10 smallest detected RTSs is in the range of $1.6\,\text{m}$ to $2.1\,\text{m}$ and can be seen as an approximation for the smallest RTS headwall heights that are detectable. Similarly, the smallest total area changes of detected RTSs are on the order of $1000\,\text{m}^2$ to $1500\,\text{m}^2$. If the size of the erosion features approaches the pixel resolution also the accuracy of the estimated volume loss increases. InSAR related processes play here the biggest role like the about 45 degree right looking viewing geometry in an ascending orbit and inaccuracies in the estimated coherence. These error sources and increased uncertainties especially for small RTSs, both in terms of spatial and vertical changes, should be considered in the interpretation and future use of the dataset."

**RC1 paragraph 6**

The results show some noise in the scaling relationship, which is certainly not unexpected given the diversity of drivers and physiographic factors that govern thaw slump development. The capacity to explain this variability based on remotely-sensed landscape factors is limited, but as stated, with further refinement of methods and proxy measures of ground conditions (ice content, soil thickness, base-level controls), there is great potential to advance our understanding of the transformations of the landscape that are underway. It would be good to see some further

speculation on the reasons for variation in the scaling exponents in different regions - what do they signify? Section 5.2 is conspicuously brief on this. In the Banks Island dataset, for instance, smaller erosion features tend to be shallow surficial failures, resulting in proportionately smaller volumes in that part of the size spectrum, and thus a steeper regression curve. In the Peel Plateau setting, there is very little confining topography to arrest headwall development, and thus the relatively larger features can get very large, again contributing to a steeper relation. Other sites may see less topographic variation across scales, which might contribute to a shallower slope on the regression curve. Glacial legacy plays a very important role in moderating this relation, as well. Some further, fairly general, geomorphic terrain interpretation could yield insights into how conditions change with scale.

**AC**

We agree that a more thorough interpretation of the observed differences in the scaling coefficient of the regression line is warranted. We added some interpretation and furthermore discuss the age component of RTSs. We added to the Discussion section 5.2:
"On the other hand, for RTSs in the Peel Plateau there is only little confining topography and deep layers of ice-rich tills which allows the headwall to grow to large sizes and consequently a steeper regression curve (Lacelle et al., 2015). The diversity in landform characteristics also contributes to the scaling relationship. In the study areas Banks Island or Noatak, shallow detachments are dominant in the small-area range. They may promote larger scaling coefficients when combined with older, deeper thaw slumps (Lewkowicz, 1987). Furthermore, most RTSs initiate as shallow active layer detachments. The gradual transition following an extreme initiation event could lead to a temporal change in the scaling coefficient. Further investigations relating the scaling coefficients to additional RTS and area characteristics (e.g. soil properties, climatic history, age of the RTSs) are needed. "

**RC1 paragraph 7**

The prospects for broad scale repeat monitoring thaw slump evolution is appealing - the work presented here shows that with a good supporting dataset of landscape information there a good possibility of achieving this, and advancing models for periglacial landscape evolution in the Anthropocene. The paper is well structured, and the charts and graphics are nicely rendered, but there are quite a few typos and grammatical issues in the text; this should be carefully reviewed before resubmission. There is some confusion regarding the numbering of figures in Section 4 and 4.1 (Figs 3-5) that require some attention. A few points are listed below. With the resolution of these minor points and a few points addressing error/uncertainty and some interpretation of the regression slopes, I recommend advancing this paper to publication.

**AC**

We carefully reviewed the manuscript for typos and grammatical errors and corrected the confusion related to the figure labelling. We corrected all listed points.

140  **Typos**

**RC1**

l. 18: suggest 'underlain' by permafrost

**RC1**

l. 29: nutrition => nutrients

145  **RC1**

l. 35: insides => insights

**RC1**

l.39: It's not clear to me that a frequency distribution is a scaling law.

**RC1**

150  l.40: 'disturbed area' => more explicitly, the erosion site (not deposition)

**RC1**

l.48 were => where

**RC1**

l.49 vertical - I think you mean horizontal, here.

155  **RC1**

l.50 were => where

**RC1**

l.56 only became available in the last few years

**RC1**

160  l.59 ..and have observed the global land mass two to three times, now.

**RC1**

l.72 ..modelling and will further improve..

**RC1**

l.83 Due to the large extent of some areas

165  **RC1**

l.95 The incident angles (?)

**RC1**

l.97 We only studied winter acquisitions due to the low..

**RC1**

170  l.144, 147 - beware affected vs effected - meaning is not clear, here.

**RC1**

l.150 volume

**RC1**

l.167 ..is to relate the area..

175  **RC1**

l.177 violine => violine. Use either comma or thin space between thousands in presentation of numbers.

**RC1**

l.184 exponential

180   **RC1**

l.185 potentially

**RC1**

l.199 By contrast, the PDF based on..

**RC1**

185   l.204 exponential decay coefficients

**RC1**

l.248 ..this universal scaling also applies to permafrost landscapes..

**RC1**

l.258 ..order of magnitude higher growth rates..

190   **RC1**

l.259 relict ice

**RC1**

l.313 ..data availability only allowed us to compute elevation changes..

**RC1**

195   l. 325 insides => insights

**Reviewer 2**

**R2**

Line 18: Only 15% of the Northern Hemisphere is underlain by permafrost. Please correct accordingly. Explanation for why one quarter is not the correct value can be found in this publication:
200 https://doi.org/10.1029/2021JF006123 . I would also suggest adding it as a reference.

**AC**

We implemented the correction and added the reference.

**RC2**

Line 54: "only remote sensing techniques are feasible": I would argue that other techniques are
205 not impossible to perform. Consider changing to "more feasible" or similar.

**AC**

We changed this part to "remote sensing techniques are the most feasible".

**RC2**

Line 55: The sentence is not clear.

210 **AC**

We omitted the term "supra-regional" and changed the sentence to: "Digital elevation models (DEMs) that cover the pan-Arctic permafrost terrain with a high enough resolution to study RTSs became only available in the last few years."

**RC2**

215 Lines 110-112: "It is difficult to assess RTS delineation procedure based only on this text. I suggest adding examples of RTS delineation from each site to the supplement."

**AC**

We added several examples of RTS delination with aditional false-color optical Sentinel-2 images to the supplement.

**RC2**

Lines 117-118: It is not clear for reader at this point what "shoreline" and "hillslope" relate to.

**AC**

We added some clarifications: "Additionally, the location of the RTS in terms of "shoreline" (located close to a waterbody) or "hillslope" (located at trenches or riverbeds) was noted."

**RC2**

Line 120: From the text, one could assume that you only delineated RTS according to the active elevation change occurring at headwall. Substantial area of RTSs is a zone where material is transported and no significant elevation change occurs.

**AC**

Yes we only delineated RTSs according to the active elevation change. In the DEM difference images only this part is clearly visible. To make this distinction more clear we specify the zone we delineate as primary scare zone and explicitly exclude the debris tongues. We added to Section 3.3 RTS attibutes the sentence:
"For all calculations we used the area outlined by the polygon indicating the areas showing an elevation change and thus a net volume loss. It is to note that this area can also be a zone of deposition, especially for small and low-relief RTSs or if the time between observations increases. Areas such as the debris tongues or zones of alluvial deposits can not be accurately detected by the DEM difference data and are not included."

**RC2**

Lines 168-177: This part would be more suitable for the methods section. Consider moving it.

**AC**

We assume that the reviewer refers to the sentence: "Due to the low density of RTSs in Yamal and Gydan and the two study regions in Taymyr we combined these to one study region (in the following "Yamal/Gydan" and "Taymyr") according to their geographical and geophysical proximity." We think that there is no clear section in the methods were we could add it and think that the decision to merge them are motivated by the results and could stay there.

**RC2**

Line 189 and other sentences describing figures before and after: It would be more appropriate
to refer to the figure inside the text, instead of writing separate sentences just for that purpose. I
would suggest extending these sentences to describe the main results shown in these figures.

**AC**

We changed the manuscript to the reviewers suggestions and rewrote and removed several sentences.

**RC2**

Line 204: Is there an established methodology on how to quantitatively discriminate megaslumps
from slumps? Otherwise I would suggest avoiding this term.

**AC**

The term "mega slump" is relatively new but has been used in several recent publications (e.g.
(Lacelle et al., 2015; Kokelj et al., 2015; Jones et al., 2019). The exact definition varies slightly
but generally includes RTSs with an area larger then $20\,ha$ $(2 \cdot 10^5\mathrm{m}^2)$.

**RC2**

Line 254: The correct would be "relict ice". Since not all of the relict ice is necessarily excess or
massive ice, using "massive" or "excess" would be more appropriate.

**AC**

We changed the term to massive ice.

**RC2**

Lines 296-297: This statement might be too general. Differences between the aspects according
to solar radiation might be relatively small in the high Arctic, where also north-facing slopes
receive quite some solar radiation during the Arctic summer. Given that RTS headwalls are close
to vertical, relatively low sun angles might still be efficient in melting ice.

**AC**

We think that the aspect factor does indeed plays an important role in RTS activity. We agree
with the reviewer that in the case of vertical headwall the impact reduces, but considering the our

study areas are located at 60-75 degree latitude and that RTSs initiate and grow at slope angles of
15-25 degree the difference in the energy availability due to solar irradiation plays an important
role. See for example Figure 3 in (Ohmura, 2012) for the hourly ration measured during summer
at Summit (Greenland) at 72.6 degree latitude.

**Typos**

**RC2**

Line 20: Missing comma after "Furthermore".

**RC2**

Line 29: You probably meant nutrient cycles.

**RC2**

Line 54: "Arctic RTSs": Are there any RTSs outside the Arctic? Consider omitting "Arctic"

**RC2**

Line 132: You probably meant "than" instead of "then".

**RC2**

Line 137: Missing comma after "For this computation"

**References**

Bernhard, P., Zwieback, S., Leinss, S., and Hajnsek, I.: Mapping Retrogressive Thaw Slumps Using Single-Pass TanDEM-X Observations, IEEE Journal of Selected Topics in Applied Earth Observations and Remote Sensing, 13, 3263–3280, https://doi.org/10.1109/JSTARS.2020.3000648, 2020.

Jones, M. K. W., Pollard, W. H., and Jones, B. M.: Rapid initialization of retrogressive thaw slumps in the Canadian high Arctic and their response to climate and terrain factors, Environmental Research Letters, 14, 055 006, 2019.

Kokelj, S., Tunnicliffe, J., Lacelle, D., Lantz, T., Chin, K., and Fraser, R.: Increased precipitation drives mega slump development and destabilization of ice-rich permafrost terrain, northwestern Canada, Global and Planetary Change, 129, 56–68, https://doi.org/https://doi.org/10.1016/j.gloplacha.2015.02.008, 2015.

Lacelle, D., Brooker, A., Fraser, R. H., and Kokelj, S. V.: Distribution and growth of thaw slumps in the Richardson Mountains-Peel Plateau region, northwestern Canada, Geomorphology, 235, 40–51, https://doi.org/10.1016/j.geomorph.2015.01.024, 2015.

Lewkowicz, A. G.: Nature and Importance of Thermokarst Processes, Sand Hills Moraine, Banks Island, Canada, Geografiska Annaler: Series A, Physical Geography, 69, 321–327, https://doi.org/10.1080/04353676.1987.11880218, 1987.

Ohmura, A.: Present status and variations in the Arctic energy balance, Polar Science, 6, 5–13, https://doi.org/https://doi.org/10.1016/j.polar.2012.03.003, special Issue: The Second International Symposium on the Arctic Research (ISAR - 2), 2012.

---

## Author Response (AR2)

Dear Peter,

thank you very much for the detailed comments to improve the manuscript. We changed the manuscript accordingly and highlight the most important changes below. Unfortunately, I found an error in some of the presented plots and data. I accidentally used the total volumetric and area change instead of the yearly change to compute the PDFs and associated rollover, cutoff and exponential decay coefficients. Thus in Figures 4, 5 and Tables S1 and S2 the presented data is based in the total volumetric and area change rates. This was due to an error when reading-in the data were the index was off by 1. Using the corrected data (yearly changes) leads to a reduction in the quantities related to the PDFs of about a factor of 4. I have corrected now Figure 4, 5, the Tables S1 and S2 and all occurrences in the text. It does not effect in any other way the content of the paper like for example the difference between study sites stay the same. I also checked all other data again and everything is correct otherwise. I highlighted all the made changes in the change document and I hope this is not a problem.

Many thanks and best regards,
Philipp Bernhard

Author Reply to the Editor comments.
In the following we appreviate: Editor Comment as EC and Author Comment as AC.

**EC:**
L5: "Please highlight in your abstract the use of PDFs for change rate distributions as I think that you may be the first (or at least among the first) to apply PDFs to RTS, so there is novelty."

**AC:** We changed the order in which we present the results in the abstract and highlight the use of PDFs. I found one study that has already presented a frequency distribution of RTSs on Svalbard. Even when in this study there was not a detailed analysis of the frequency distribution I don't think it is fair to say that we are the first. I cited the paper in the introduction.

**EC:**
L18-20: "Too sensational. Most thaw from a regional perspective will not be rapid, but will be gradual, "Press" thaw. Only specific locations will experience faster thaw, usually where there is excess ground ice and thermokarst. Thermokarst operates on up-to decadal scales, so it that "rapid"? See Grosse et al. 2011) who say "Press disturbances of relatively slow but persistent nature such as top-down thawing of permafrost, and changes in hydrology, microbiological communities, pedological processes, and vegetation types, as well as pulse disturbances of relatively rapid and local nature such as wildfires and thermokarst,"
RTS are rapid to initiate, a "Pulse" thaw", but the thaw continues for some time until the RTS stabilizes.
I think that you could easily re-cast this and rely on the "Press" and "Pulse" language presented in Grosse et al. 2011, and use as "thermokarst" instead of

"rapid thaw". These alternative terms, in my mind, are more effective at describing the various thaw rates and will get your point across better to geomorphologists."

L20: "Sensational. All permafrost has these impacts, not just Pulse type thaw."

L23: "Again, the thaw event presented by an RTS is not rapid, it is the initiation that is "rapid""

**AC:**
We agree with the assessment and we changed the part in the introduction and used the suggested terms "Press" and "Puls" disturbances.

"With climate warming these permafrost regions become increasingly vulnerable to thaw. This thaw manifest itself first in a slow but gradual deepening of the seasonally thawed active layer (press disturbances) and secondly in a more rapid and local way by the development of thermokarst features (pules disturbances) (Grosse et al., 2011; Schuur et al., 2015). Both forms of permafrost degradation have major impacts by changing ecosystem and hydrological equilibria and impact the Earth system on a global scale by reinforcing climate change with the additional mobilization of organic carbon that was previously stored in the frozen soil. One important thermokarst feature arising from pulse disturbances are retrogressive thaw slumps (RTS)."

**EC:**
L36,37: Please highlight if you are the first to apply PDF to RTS.
**AC:**
We added to the manuscript: "Currently there is only one study quantifying the area frequency distributions of RTSs, were orthophotos for a study site on Svalbard was used to measure the area disturbed by RTSs (Nicu et al., 2021)."
Nicu, I. C., Lombardo, L., and Rubensdotter, L.: Preliminary assessment of thaw slump hazard to Arctic cultural heritage in Nordenskiöld Land, Svalbard, Landslides, pp. 1–13, 2021.

**EC:**
L 73: "For consistency, please use either "sites" or "regions" throughout. I suggest "sites" as you use "regions" in other contexts."
**AC:**
We use now the term "study site" and changed all occurances in the manuscript

**EC:**
L83 "Not clear. Do you mean: "Within these extensive regions we selected representative locations for our study sites"?"
**AC:**
Yes, thats what we mean and we adopted the suggested fomulation.

**EC:**
L86.87: "Topography and soil type are almost always drivers of near-surface ground ice variation at only at very local scales, and on the order of metres. At larger scales, there are typically much different reasons for high and extensive ground ice contents that yield RTS. E.g., buried glacial ice, or massive syngenetic ice, or syngenetic permafrost aggradation as in Yedoma deposits. There may be relations between topography and soil type with RTS initiation,

but the ground ice that you are really talking about, which relates to RTS, is not *due to* topography.
Please clarify this section for the reader."
**AC:**
We clearified this distinction between large and small scale variation in ground ice. We added to the manuscript:
"On large scales, high ground ice content is associated with the climatic history (e.g. syngenetic ice-wedges) and the associated extent of past glacial ice (e.g. buried glacial ice). On small scales ground ice content can vary due to for example soil type (Lacelle et al., 2004).

**EC:**
L93: Tables have single sentence titles, rather than captions as figures do. Please move this table title above the table. Please repeat for other tables.
**AC:**
We corrected the formating for all tables

**EC:**
L115-121:
"This paragraph needs an introductory sentence, probably something that links to the final sentence."
"If there are any please cite, otherwise there are none rather than a limited number."
"Are these limits based on using TanDEM-X DEM, or from some other source? If related to the TanDEM-X DEM, then can't you simply say that these reflect the accuracy and precision of the TanDEM-X DEM?
It is just not clear in this section if you are talking about inventories, or your own data."
"Please re-write for clarity. What is the role played?"
**AC:**
To adress the made points we rewrote the part in the manuscript:
" The error sources and uncertainties that govern the lower RTS detection limit in terms of headwall height and retreat rate are manifold and difficult to quantify. This is mainly due to the small amount of available high resolution, three dimensional RTS inventories (Swanson and Nolan, 2018; Van der Sluijs et al., 2018), were also timescales on which the RTSs are monitored     plays an important role. To get an estimate on the lower limit of RTS induced elevation changes to be detectable we can analyse the smallest detected RTSs in our sample. The 10 smallest detected RTSs have elevation changes in the range of 1.6 - 1.9 m and can be seen as an approximation for the smallest RTS headwall heights that are detectable, which are on the same order then the general TanDEM-X DEM accuracies. Similarly, the smallest total area changes of detected RTSs are on the order of 500 - 1000m$^2$,corresponding to about 10 - 12 pixels. Consequently, if the size of the erosion features approaches the pixel resolution also the accuracy of the estimated volume loss increases. Additionally, processes related to the observation properties and interferometric processing further complicate the error estimations. For example the about 40 degree right looking viewing geometry leads to different pixels resolution depending on aspect and slope of the observed area.
These error sources and increased uncertainties especially for small RTSs, both in terms of spatial and vertical changes, should be considered in the interpretation and future use of the dataset.

**EC:**
L133: "This doesn't quite make sense to me. Do you mean: " For some study sites (list them) several winters of observations were available (2010/11, 2011/12, and 2012/13)."
**AC:**
We mean that parts of the study sites have not a complete coverage in all of these winters. This is the case for all sites. To clearify we slightly changed the sentence to:
"For parts of the study sites observations during winters in 2010/11, 2011/12 and/or 2012/13 were available."

**EC:**
Table 2: "Please always use names as indicated in Tables and figures for consistency. Please check the entire manuscript for consistency"
**AC:**
We checked the manuscript again and always use the study site abbreviations.

**EC:**
Figure 5: "For consistency, please write either "(a)" or "a)", but not both. Currently both styles are used throughout the text body, captions, and figures."
**AC:**
We changed all labeling to a), b)...

**EC:**
L220: " In Methods Section 3.5 the scaling parameter is given as gamma, but in this section is given as alpha. Please be consistent. Depending on what you use, you may have to adjust details in figures or tables."
**AC:**
We use now alpha throughout the manuscript for the volume-to-area scaling coefficent.

**EC:**
Figure 7: "Please clarify if the centerline values and the mean/median are for the entire study region.
Rather than splitting the violin plot, as the probability density distributions are often quite different for hillslope versus shoreline, it might be better to have two violin plots for each study region according to location."
**AC:**
We clarified that the white dot and the thick line are related to the RTSs in the total study site. We think that this presentation (top shoreline and bottom hillsope) makes the comparison easy.We think that adding additional 8 violin plots makes the Figure more confusing and does not add important additional information.

**EC:**
L.286"I'm not sure that most would consider ground ice content as a soil property, as ground ice is most common below the soil (ground ice that matters to RTS in any case). Perhaps: "physical characteristics of ground materials"?"
**AC**
We added the suggestions.

**EC:**
L303: "transition to what? Deeper thaw and RTS development? Please clarify."
**AC:**
We change the sentence to: "Furthermore, most RTSs initiate as shallow active layer detachments. The gradual increase in headwall heights following the initiation event could lead to a temporal change in the scaling coefficient."

---

## Editor Decision (ED2)

[revised manuscript text omitted]

**Figure S 1.** Example of RTS polygones in the Chukotka study region located at N65.93 W-178.82. Left: DEM difference image between winter 2010/11 and 2016/17. Right: False color Sentinel-2 image taken on  09.09.2016.

[Figure]

**Figure S 2.** Example of RTS polygones in the Tuktoyaktuk study region located at N69.08 W-134.02. The black arrows indicate the RTS lcoations Left: DEM difference image between winter 2010/11 and 2016/17. Right: False color Sentinel-2 image taken on the 06.08.2016.

[Figure]

**Figure S 3.** Example of RTS polygones in the Peel study region located at N67.26 W-135.27. Left: DEM difference image between winter 2010/11 and 2016/17. Right: False color Sentinel-2 image taken on  06.08.2016.

[Figure]

**Figure S 4.** Example of RTS polygones in the Yamal study region located at N71.09 W70.40 . Left: DEM difference image between winter 2010/11 and 2016/17. Right: Fasle color Sentinel-2 image taken on  12.08.2016.

[Figure]

**Figure S 5.** Correlation coefficients between all computed quantities of all areas. A value below -0.64 and above 0.64 are statistically significant (t-Test with a p-value < 0.05.

[Figure]

**Figure S 6.** Area to Volume scaling for each study area.

[Figure]

**Figure S 7.** Aspect distribution of all study areas.

---

## Author Response (AR3)

Dear Peter,

thanks again for the comments to improve the manuscript. We changed the manuscript accordingly and changed the tense to active where appropriate. I highlighted all the made changes in the change document.

Many thanks and best regards,
Philipp Bernhard

---

## Author Response (AR4)

Dear Peter,

thanks again for the helpful comments thought the review process We changed the manuscript according to your last comments and additionally corrected a mistake in a citation.

Many thanks and best regards,
Philipp Bernhard